# A Novel Weighted Block Sparse DOA Estimation Based on Signal Subspace under Unknown Mutual Coupling

Yulong Liu [1,2], Yingzeng Yin [1,*], Hongmin Lu [1,*] and Kuan Tong [2]

1   School of Electronic Engineering, Xidian University, Xi'an 710071, China; lyl22601@163.com
2   China Research Institute of Radiowave Propagation, Qingdao 266107, China; tongkuan2020@163.com
*   Correspondence: yyzeng@mail.xidian.edu.cn (Y.Y.); hmlu@mail.xidian.edu.cn (H.L.)

**Abstract:** In this paper, a novel weighted block sparse method based on the signal subspace is proposed to realize the Direction-of-Arrival (DOA) estimation under unknown mutual coupling in the uniform linear array. Firstly, the signal subspace is obtained by decomposing the eigenvalues of the sampling covariance matrix. Then, a block sparse model is established based on the deformation of the product of the mutual coupling matrix and the steering vector. Secondly, a suitable set of weighted coefficients is calculated to enhance sparsity. Finally, the optimization problem is transformed into a second-order cone (SOC) problem and solved. Compared with other algorithms, the simulation results of this paper have better performance on DOA accuracy estimation.

**Keywords:** weighted block sparse; signal subspace; mutual coupling; DOA

## 1. Introduction

DOA estimation is widely used in radar [1], communication [2], sonar [3], and other fields [4–6], which is one of the most important topics in array signal processing [7]. In the past decades, there has been rapid progress, such as the multiple signal classification (MUSIC) algorithm [8], the estimation of signal parameters via the rotational invariant technique (ESPRIT) algorithm [9], the maximum likelihood algorithm (ML) [10], and so on. However, in the real environment, there are many non-ideal factors such as gain and phase error [11], mutual coupling error [12], etc., which lead to the deviation between the actual steering vector and the ideal steering vector of the received signal [13]. This deviation seriously affects the accuracy of DOA estimation [14], and consequently, the performance of the above algorithms is seriously affected [15].

In order to solve these problems, Ye in [16] proposes a preprocessing algorithm that sacrifices the array aperture to suppress the negative influence of mutual coupling between array elements [17]. After preprocessing, the array steering vector can be equivalent to the ideal Vandermonde vector form, and the direction-finding performance is obviously improved [18]. Dai in [19] uses the proposed preprocessing method of [16] to achieve coherent signal DOA estimation, but the essence of this method is still array smoothing technology, and preprocessing and smoothing result in a significant loss of array aperture. Liao in [20] proposes a method for estimating DOA under unknown mutual coupling using the rank deficiency method. After obtaining DOA, we reverse calculate the mutual coupling error coefficient, which opens up new ideas for research. Li in [21] simultaneous preprocessing technology reduces mutual coupling, and the ESPRIT algorithm achieves DOA estimation. In recent years, signal sparsity theory has been widely applied in the field of array signal processing. Compared with the algorithms based on the subspace decomposition technique, the DOA estimation algorithms based on sparse theory require a smaller number of snapshots and have better performance at low SNR [22]. In addition, Bilik in [23] explained the rationality of using sparse theory for array signal processing. Dai in [24] uses the proposed preprocessing method of [16] to achieve ideal steering vector

that has a Vandermonde matrix structure, and a block sparse structure was constructed to achieve DOA estimation, but there is still room for improvement in the performance of this method. Wang in [25] was the first to propose using block sparsity to achieve DOA estimation under unknown mutual coupling, providing an important theoretical reference for researching DOA estimation under array mutual coupling. However, facing different parameters, this algorithm may not always be able to successfully estimate the DOA of the incident signal, so there is still significant room for improvement in stability methods. Zhang in [26] and Tang in [27] achieve DOA estimation under unknown mutual coupling from the perspective of numerical iteration while also achieving relatively ideal performance. Dai in [28] has achieved real-value DOA estimation under array mutual coupling error, reduced computational complexity, and achieved relatively ideal DOA estimation performance. Meng in [29] proposes a sparse solution method under the coexistence of signal coherence and mutual coupling, but the algorithm will have different performance when facing different parameters, so it still needs to be improved. Meng in [30] proposes a novel block sparse estimation DOA method based on weighted signal subspace, and Antonello in [31] proposes a lightweight technique for artifact correction and compensation in DOA Estimation, which provides us with a new perspective.

To sum up, whether it's in classical theory or sparse signal processing, significant progress has been made in mutual coupling DOA estimation. However, there is still room for improvement in DOA estimation techniques [32]. The purpose of this paper is to realize the DOA estimation under unknown mutual coupling based on signal subspace. Different from traditional sparse DOA estimation methods, our proposed method utilizes the specific deformation of the production of MCM and steering vector to construct a novel block sparse model and seeks a set of weighted coefficients to enhance sparsity, thereby achieving better DOA estimation performance. For specific comparisons, please refer to Section 4.

The arrangement of this paper is as follows: in Section 2, the received signal model and MUSIC algorithm under mutual coupling are introduced. In Section 3, a novel weighted block sparse DOA estimation method based on signal subspace is realized. In Section 4, simulation results are given, and the advantages of our proposed algorithm are illustrated by simulation effectiveness. Finally, in Section 5, conclusions are given.

## 2. Data Model

### 2.1. Signal Model

As shown in Figure 1, assuming that $Q$ far-field narrowband signals are incident on a uniform linear array composed of $M$ array elements with a spacing of $\lambda/2$, the array received snapshot data at time $t$ is

$$X(t) = \sum_{q=1}^{Q} s_q(t) C a(\theta_q) + n(t) = CAS(t) + n(t) \tag{1}$$

where, $X(t) \in \mathbb{C}^M$ represents the received snapshot data at time $t$ that contains noise, $a(\theta_q) = [1, e^{-j\frac{2\pi d \sin \theta_q}{\lambda}}, \ldots, e^{-j\frac{2\pi(M-1)d \sin \theta_q}{\lambda}}]^T \in \mathbb{C}^M$ is the ideal steering vector of the $q$-th incident signal, $\lambda$ is the signal wavelength, $[\cdot]^T$ is transpose operation, $A = [a(\theta_1), a(\theta_2), \ldots, a(\theta_Q)] \in \mathbb{C}^{M \times Q}$ is the ideal array manifold matrix, $S(t) \in \mathbb{C}^Q$ is the source signal, $n(t)$ represents Gaussian additive white noise $\sim \mathcal{CN}(0, \sigma_n^2 I_M)$, $\sigma_n^2$ is the noise variance, signal and noise are independent of each other, $C$ represents the mutual coupling matrix (MCM) composed of mutual coupling coefficients, $C = \text{Toeplitz}([c, \mathbf{0}_{M-p}^T])$, $C \in \mathbb{C}^{M \times M}$, where $c = [1, c_1, \ldots, c_{p-1}]$, and satisfying $1 > |c_1| > |c_2| > \cdots > |c_{p-1}|$. Taking into account the influence of mutual coupling, the received signal can be represented as

$$X(t) = \widetilde{A}\beta s(t) + n(t), \quad \widetilde{A} = [Ca(\theta_1), \ldots, Ca(\theta_Q)] \in \mathbb{C}^{M \times Q} \tag{2}$$

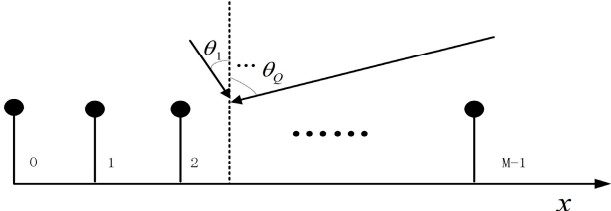

**Figure 1.** Uniform linear array.

*2.2. Eigenvalue Decomposition and MUSIC Algorithm*

Since signal and noise are assumed to be independent and uncorrelated, the covariance matrix can be expressed as

$$
\begin{aligned}
\boldsymbol{R} &= \mathrm{E}\left\{\boldsymbol{X}(t)\boldsymbol{X}^H(t)\right\} \\
&= \boldsymbol{C}\boldsymbol{A}\mathrm{E}\left\{\boldsymbol{s}(t)\boldsymbol{s}^H(t)\right\}\boldsymbol{A}^H\boldsymbol{C}^H + \sigma_n^2\boldsymbol{I}_N \\
&= \boldsymbol{C}\boldsymbol{A}\boldsymbol{R}_s\boldsymbol{A}^H\boldsymbol{C}^H + \boldsymbol{R}_N
\end{aligned}
\tag{3}
$$

In Formula (3), $\boldsymbol{R}_s = E\left\{\boldsymbol{s}(t)\boldsymbol{s}^H(t)\right\}$ is signal correlation matrix, $\boldsymbol{R}_N = \sigma_n^2\boldsymbol{I}_N$ is noise correlation matrix, and eigenvalue decomposition processing is performed on $\boldsymbol{R}$

$$
\begin{aligned}
\boldsymbol{R} &= \sum_{q=1}^{Q}\lambda_q\boldsymbol{u}_q\boldsymbol{u}_q^H + \sum_{q=Q+1}^{M}\sigma_q^2\boldsymbol{u}_q\boldsymbol{u}_q^H \\
&= \boldsymbol{U}_s\boldsymbol{\Sigma}_s\boldsymbol{U}_s^H + \boldsymbol{U}_n\boldsymbol{\Sigma}_n\boldsymbol{U}_n^H
\end{aligned}
\tag{4}
$$

After $\boldsymbol{R}$ eigenvalue decomposition, eigenvalues are divided into *M-Q* small eigenvalues and Q large eigenvalues. In Formula (4), $\boldsymbol{U}_s = \mathrm{span}\left\{\boldsymbol{u}_q, q = 1, 2, \cdots, Q\right\} \in \mathbb{C}^{M\times Q}$ denotes the signal subspace composed of eigenvectors corresponding to Q large eigenvalues, $\boldsymbol{U}_n = \mathrm{span}\left\{\boldsymbol{u}_q, q = Q+1, Q+2, \cdots, M\right\} \in \mathbb{C}^{M\times(M-Q)}$ denotes the noise subspace constructed by eigenvectors corresponding to *M-Q* small eigenvalues, and because the steering vector is orthogonal to the noise subspace [8], it can be expressed as

$$
\mathrm{span}\left\{\boldsymbol{u}_q, q = Q+1, Q+2, \ldots, M\right\} \perp \mathrm{span}\left\{\boldsymbol{C}\boldsymbol{a}(\theta_q), q = 1, 2, \ldots, Q\right\}
\tag{5}
$$

Simplified Formula (5), we can get

$$
(\boldsymbol{C}\boldsymbol{a}(\theta_q))^H\boldsymbol{U}_n = 0
\tag{6}
$$

Without unknown mutual coupling errors, the spatial spectrum estimation of the MUSIC algorithm is

$$
P_{MUSIC} = \frac{1}{\boldsymbol{a}^H(\theta)\boldsymbol{U}_n\boldsymbol{U}_n^H\boldsymbol{a}(\theta)}
\tag{7}
$$

In practical applications, the theoretical covariance matrix is often replaced by a sampling covariance matrix, which can be expressed as

$$
\begin{aligned}
\hat{\boldsymbol{R}} &= \frac{1}{K}\sum_{t=1}^{K}\boldsymbol{X}(t)\boldsymbol{X}^H(t) \\
&= \hat{\boldsymbol{U}}_s\hat{\boldsymbol{\Sigma}}_s\hat{\boldsymbol{U}}_s^H + \hat{\boldsymbol{U}}_n\hat{\boldsymbol{\Sigma}}_n\hat{\boldsymbol{U}}_n^H
\end{aligned}
\tag{8}
$$

Due to the influence of finite samples, both $\hat{\boldsymbol{U}}_s$ and $\hat{\boldsymbol{U}}_n$ have certain errors, which leads to the fact that the steering vector is not completely orthogonal to the noise subspace. Therefore, the maximized power peak spectrum search is generally used to complete DOA estimation. However, not only the number of sampling snapshots is limited, but also the mutual coupling matrix $\boldsymbol{C}$ seriously affects the orthogonality of the signal steering

vector and the noise subspace. Therefore, the spatial spectrum under mutual coupling is specifically expressed as:

$$\hat{P}_{MUSIC} = \frac{1}{a^H(\theta)C^H\hat{U}_n\hat{U}_n^H Ca(\theta)} \tag{9}$$

However, $C$ is unknown, the performance of the music algorithm has significantly decreased.

## 3. Proposed Method

### 3.1. The Connection between the Signal Subspace and the Array Manifold Matrix

According to [33], the array manifold matrix and signal subspace span into the same subspace, and they have a certain mathematical connection since

$$U_s U_s^H + U_n U_n^H = I_M, U_s^H U_s = I_Q \tag{10}$$

According to Formula (3), we can get

$$\begin{aligned} CAR_s A^H C^H &= R - \sigma_n^2 I_M \\ &= U_s \Sigma_s U_s^H + \sigma_n^2 U_n U_n^H - \sigma_n^2 I_M \\ &= U_s \left( \Sigma_s - \sigma_n^2 I_M \right) U_s^H \end{aligned} \tag{11}$$

Right-multiplying both sides of the equation by $U_s$, we can obtain

$$U_s = CAR_s A^H C^H U_s \left( \Sigma_s - \sigma_n^2 I_M \right)^{-1} = CAB = \widetilde{A}B \tag{12}$$

where $B = R_s A^H C^H U_s \left( \Sigma_s - \sigma_n^2 I_M \right)^{-1} \in \mathbb{C}^{Q \times Q}$. The Formula (12) is an important mathematical expression that bridges the gap between the signal subspace and the array manifold matrix.

### 3.2. Novel Weighted Block Sparse DOA Estimation

The mutual coupling matrix $C$ is unknown, fortunately, according to the special structure of MCM and steering vector, $Ca(\theta_q)$ can be represented as

$$Ca(\theta_q) = H(\theta_q)c \tag{13}$$

where $c = [1, c_1, \ldots, c_{p-1}]^T \in \mathbb{C}^{p \times 1}$ is the mutual coupling coefficient, $H(\theta_q) = [E_1 a(\theta_q), \ldots, E_p a(\theta_q)] \in \mathbb{C}^{M \times p}$ is the special deformation of $a(\theta_q)$, and the number of its column vectors is related to the number of mutual coupling coefficients, $E_m a(\theta_q)$ is m-th column of $H(\theta_q)$, where $E_m$ is defined as

$$[E_m]_{ij} = \begin{cases} 1, \text{if}[C]_{ij} = [c]_m \\ 0, \text{otherwise} \end{cases} \tag{14}$$

$[c]_m$ represents the $m$-th element of $c$, $[C]_{ij}$ represents the element in the $i$-th row and $j$-th column of $C$, $[E_m]_{ij}$ represents the element in the $i$-th row and $j$-th column of $E_m$. Obviously, the number of $H(\theta_q)$ column vectors depend on the number of mutual coupling coefficients. Therefore, with the help of a special form of $Ca(\theta_q)$ and $U_s$, $U_s$ can be deformed as [34]

$$U_s = CAB = \widetilde{H}(\theta)(B \otimes c) \tag{15}$$

where $\widetilde{H}(\theta) = [H(\theta_1), \ldots, H(\theta_Q)] \in \mathbb{C}^{M \times Qp}$, $B \otimes c \in \mathbb{C}^{Qp \times Q}$ is block signal after special deformation, which is related to the mutual coupling coefficient and signal $B$. Formula (15) shows that DOA information is required for the establishment of $\widetilde{H}(\theta)$.

Due to the sparse model of the DOA estimation problem in the spatial domain, a sparse model can be established to solve this problem, assuming that the grid points are set as follows $\Theta = [\theta_1, \ldots, \theta_G]$. $\boldsymbol{H}(\theta_q)$ is not a column vector, so a block sparse model is established to solve the DOA estimation problem under the unknown mutual coupling. In order to reconstruct the sparse signal space spectrum from Formula (15), we need to construct a new over-complete matrix $\bar{\boldsymbol{H}}(\theta)$ that contains all possible DOA.

$$\bar{\boldsymbol{H}}(\theta) = [\boldsymbol{H}(\theta_1), \ldots, \boldsymbol{H}(\theta_G)] \in \mathbb{C}^{M \times Gp} \tag{16}$$

$G$ represents the number of grid points in the over-complete set, $G \gg M > Q$. Therefore, in the sparse model, $\boldsymbol{U}_s$ can be represented as

$$\boldsymbol{U}_s = \boldsymbol{CAB} = \bar{\boldsymbol{H}}(\theta)(\bar{\boldsymbol{B}} \otimes \boldsymbol{c}) = \bar{\boldsymbol{H}}(\theta)\widehat{\boldsymbol{B}}_\theta \tag{17}$$

$\bar{\boldsymbol{B}} \in \mathbb{C}^{G \times Q}$ is the mathematical model based on $\boldsymbol{B}$ under the sparse model, most of its elements are 0 elements. Whether a certain row of elements in $\bar{\boldsymbol{B}}$ is all zero depends on whether there is a signal incident at the corresponding DOA. $\widehat{\boldsymbol{B}}_\theta = (\bar{\boldsymbol{B}} \otimes \boldsymbol{c}) \in \mathbb{C}^{Gp \times Q}$, at this point, whether the elements in the $(p(g-1)+1)$−th to $pg$-th rows of matrix $\widehat{\boldsymbol{B}}_\theta$ are all 0 depends on whether the elements in the $g$-th row of $\bar{\boldsymbol{B}}$ are all 0, $g = 1, \ldots, G$.

$\bar{\boldsymbol{H}}(\theta)$ can be established based on the angles corresponding to grid points in the overcomplete set, while $\widehat{\boldsymbol{B}}$ is unknown. To recover the block sparse matrix $\widehat{\boldsymbol{B}}$, a block sparse optimization model is established. In this model, the $l_0$ norm penalty is selected as an ideal measure of sparsity, and considering it as the objective function, considering the constraint function, the corresponding optimization problem can be specifically represented as

$$\min \| \widehat{\boldsymbol{B}}_\theta^{l_F} \|_0 \text{ s.t. } \| \boldsymbol{U}_s - \bar{\boldsymbol{H}}(\theta)\widehat{\boldsymbol{B}}_\theta \|_F \leq \xi \tag{18}$$

where $\xi$ is regularization parameter, which controls the upper bound of the fitting error.

$\widehat{\boldsymbol{B}}_\theta^{l_F} = [\widehat{\boldsymbol{B}}_{\theta_1}^{l_F}, \widehat{\boldsymbol{B}}_{\theta_2}^{l_F}, \cdots, \widehat{\boldsymbol{B}}_{\theta_G}^{l_F}]^{\mathrm{T}} \in \mathbb{R}^G$, and $\widehat{\boldsymbol{B}}_{\theta_g}^{l_F}$ is the $l_F$ norm value of $\widehat{\boldsymbol{B}}_{\theta_g}$, $\widehat{\boldsymbol{B}}_{\theta_g}$ is $g$-th block of $\widehat{\boldsymbol{B}}_\theta$, which is composed of the $(p(g-1)+1)$-th to the $pg$-th rows of $\widehat{\boldsymbol{B}}_\theta$. Since $\widehat{\boldsymbol{B}}_\theta$ is block sparse signal, after recovering signal $\widehat{\boldsymbol{B}}_\theta$, finding the top Q largest $\widehat{\boldsymbol{B}}_{\theta_g}^{l_F}$ can obtain DOA. $l_0$ norm constrained scheme can achieve the optimal recovery performance, however, the $l_0$ norm problem is an Non-deterministic Polynomial (NP) hard problem that is extremely difficult to solve in mathematical theory. To make the problem solvable, the $l_1$ norm is used instead of the $l_0$ norm to relax the constraint conditions. At this point, the optimization problem can be expressed as

$$\min \| \widehat{\boldsymbol{B}}_\theta^{l_F} \|_1 \text{ s.t. } \| \boldsymbol{U}_s - \bar{\boldsymbol{H}}(\theta)\widehat{\boldsymbol{B}}_\theta \|_F \leq \xi \tag{19}$$

Formula (19) is a convex optimization problem; however, the $l_1$ norm affects the performance of sparse recovery, so the solution result is not always ideal. In order to enhance the sparsity of the solution, a weighted matrix was designed for the weighted $l_1$ norm minimization problem to obtain a more ideal solution, which improves the accuracy of DOA estimation. In order to enforce the sparsity of the solution and the performance of DOA estimation, a weighted matrix based on the orthogonality between the signal steering vector and the noise subspace is designed. The product of the steering vector and the noise subspace can be expressed as

$$\delta = \boldsymbol{a}^H(\theta_g)\boldsymbol{C}^H\boldsymbol{U}_n\boldsymbol{U}_n^H\boldsymbol{Ca}(\theta_g) \tag{20}$$

Transform Formula (20), we can obtain

$$\delta = c^H Q(\theta_g) c \tag{21}$$

where $Q(\theta_g) = H^H(\theta_g) U_n U_n^H H(\theta_g) \in \mathbb{C}^{p \times p}$, the rank of $H(\theta_g)$ is $p$, $g = 1, \ldots, G$, the rank of $U_n U_n^H$ is $M - Q$. When $M - Q \geq p$, rank $Q(\theta_q) = p$. However, when $\theta_g = \theta_q$, that is, when the angle corresponding to the grid point is equal to the DOA of the incident signal, according to Formula (6), we can get that

$$c^H Q(\theta_q) c = a^H(\theta_q) C^H U_n U_n^H C a(\theta_q) = 0 \tag{22}$$

$Q(\theta_q) = H^H(\theta_q) U_n U_n^H H(\theta_q) \in \mathbb{C}^{p \times p}$, $Q(\theta_q)$ is no longer full rank [35], therefore, the determinant of $Q(\theta_q)$ is 0, i.e.,

$$\det[Q(\theta_q)] = 0 \tag{23}$$

where det $[\cdot]$ stands for the determinant of a matrix. In summary, the matrix of $Q(\theta_g)$ only exhibits rank deficiencies at specific grid points. Therefore, the corresponding values are calculated as weighted values to enhance sparsity, which is based on the orthogonality principle between the steering vector and noisy subspaces. The weighted value is defined as

$$\widetilde{w}_g = \det[Q(\theta_g)] = \det[H^H(\theta_g) U_n U_n^H H(\theta_g)] \tag{24}$$

Utilizing the $\widetilde{w}_g$, a weighted matrix is defined as

$$\mathbf{W} = \mathrm{diag}\{\mathbf{W}\} \tag{25}$$

where diag$\{\cdot\}$ represents taking a diagonal matrix, $\mathbf{W} = [w_1, w_2, \cdots, w_G]$, $w_g = \widetilde{w}_g / \max\{\widetilde{w}_1, \widetilde{w}_2, \cdots, \widetilde{w}_G\}$. In the optimization problem of $l_1$ norm, the small entries of $\widehat{B}_\theta^{l_F}$, who are more closer to zero, are punished by large weights, and those larger entries of $\widehat{B}_\theta^{l_F}$ are reserved by small weights. Then a reweighted optimization model based on $l_1$-norm is given as follows:

$$\min \parallel \mathbf{W} \widehat{B}_\theta^{l_F} \parallel_1 \ \text{s.t.} \ \parallel U_s - \bar{H}(\theta) \widehat{B}_\theta \parallel_F^2 \leq \xi^2 \tag{26}$$

In order to better solve this optimization problem, the second-order cone (SOC) model can be established as

$$\min_{\delta, r, \widehat{B}_\theta} \delta, \ \text{s.t.} \mathbf{1}^T r \leq \delta, \widehat{B}_{\theta_g}^{l_F} \leq r_g \parallel U_s - \bar{H}(\theta) \widehat{B}_\theta \parallel_F \leq \xi, \ g = 1, 2, \cdots, G \tag{27}$$

where $\delta$ is the objective function and is a real number, $r = [r_1, r_2, \ldots, r_G]^T$, $\mathbf{1}^T$ is a row vector that is all 1, $\mathbf{1}^T \in \mathbb{R}^{1 \times G}$. The regularization parameter $\xi^2$ has been suggested to choose the upper bound of $\parallel U_s - \bar{H}(\theta) \widehat{B}_\theta \parallel_F^2$ with 99% confidence interval. The optimization problem (27) can be solved by using the convex optimization toolbox CVX in MATLAB R2023a.

## 4. Simulation

This paper includes simulation experiments as follows: comparison of the weighted and unweighted spatial spectrum, comparison of DOA estimation accuracy under different input signal-to-noise ratios (SNR), comparison of DOA estimation accuracy under different input snapshots; comparison of the probability of resolution (PR) under different input SNR; and comparison of the probability of resolution under different input snapshots. All simulation environments are MATLAB R2023a with an Intel Core i7-13620H, 2.40 GHz processor with 16 GB of memory.

The first experiment is to demonstrate that the weighted spatial spectrum is sharper and can improve the accuracy of DOA estimation, assuming that there is a uniform linear array composed of 9 array elements with a spacing of half a wavelength, i.e., M = 9. There are two far-field narrowband signals incident from $\theta_1 = -30.1°$ and $\theta_2 = 40°$, respectively. The initial radio frequencies were set to 0.301 GHz and 0.305 GHz. After the down-conversion stage, the signal frequencies were 1 MHz and 5 MHz. The number of sampling snapshots is fixed at K = 200, and the input SNR = 10 dB. The number of grid points is 181 with an interval is 1°, that is, G = 181. In the mutual coupling error coefficient matrix $c = [1, 0.684 + 0.563i]$, it should be noted that the mutual coupling errors of all experiments in this paper are the same as those mentioned above.

As shown in Figure 2, the red line represents the weighted spatial spectrum obtained by solving Formula (27); the black lines are the unweighted spatial spectrum obtained by solving Formula (19). We can observe that the weighted peak spectrum is sharper, thus having better resolution, and the spectrum peaks correspond to the DOA of the incident signal to be solved, thus having higher DOA estimation accuracy.

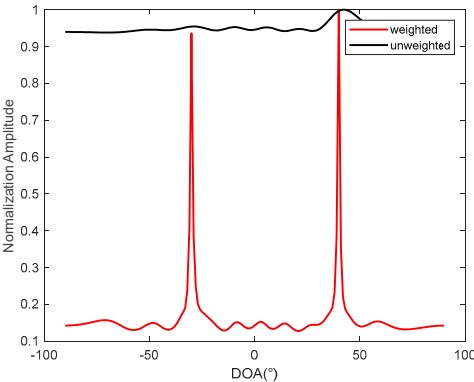

**Figure 2.** Weighted and unweighted spatial spectrum.

The second experiment is in order to illustrate the advantages of the proposed algorithm in terms of the accuracy of DOA under different SNRs: set the simulation parameters as follows: the number of uniform linear array elements M = 9, there are two far-field narrowband signals that are incident from $\theta_1 = -30.1°$ and $\theta_2 = 49°$, respectively, and the signals are uncorrelated with each other, i.e., $Q = 2$. The two signal powers are equal, the number of sampling snapshots of the two signals is K = 200, and the input SNR is 0~15 dB, with an interval of 5 dB. The mutual coupling coefficient is consistent with the first experiment. we compare it with other DOA estimation algorithms, i.e., select sparse representation (SR) method [24], block sparse representation (BSR) method [25], robust weighted subspace (RWS) method [30], mutual coupling special deformation (MCSD) method [20], mutual coupling preprocessing (MCP) method [16], mutual coupling based on the ESPRIT (MCE) method [21] and Cramér–Rao Bound (CRB) [36].

The root mean square error of DOA is used to quantitatively analyze the accuracy of DOA estimation. The expression is:

$$\text{RMSE}_\theta = \sqrt{\frac{1}{NQ} \sum_{n=1}^{N} \sum_{q=1}^{Q} \left( \theta_q - \hat{\theta}_{n,q} \right)^2} \tag{28}$$

$N$ represents the number of Monte Carlo experiments, all experiments in this paper, $N = 200$. $\hat{\theta}_{n,q}$ denotes the calculated $\theta_q$ by the $n$-th Monte Carlo experiment.

As shown in Figure 3, the performance of methods SR, MCP and MEC is not as good as the proposed methods, possibly because these methods are based on pre-processing, which results in a certain loss of array aperture. The performance of methods BSR and SR is not ideal, which may be because the sparsity of the block sparse model established by these methods is not ideal. Compared with the other algorithms, the DOA estimation

performance of our proposed algorithm has the minimum RMSE of DOA estimation in different SNRs. Especially at low signal-to-noise ratios, the performance of the proposed algorithm is significantly superior to other performance factors. This is because, after eigenvalue decomposition, the noisy subspace has been removed, and the algorithm is less affected by noise. Starting at 5 dB, the DOA estimation error of the proposed algorithm no longer changes. This is because DOA estimation is affected by the grid point setting. If the incident signal DOA is not at the corresponding DOA of the grid point, there will inevitably be errors. The accuracy of sparse DOA estimation is affected by the grid spacing.

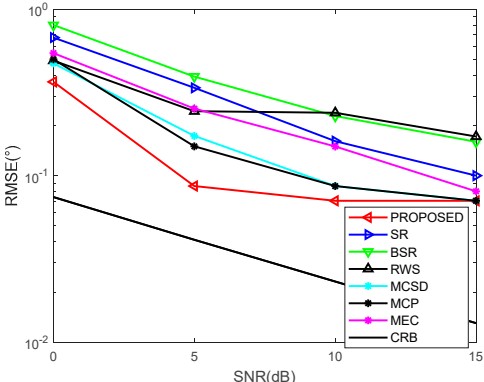

**Figure 3.** The DOA estimation performance with different SNR.

The third experiment is in order to illustrate the advantages of the proposed algorithm in terms of the accuracy of DOA under different snapshots: set the simulation parameters as follows: the number of uniform linear array elements M = 9, there are two far-field narrowband signals of equal power that are incident from $\theta_1 = -30.1°$, $\theta_2 = 49°$, $Q = 2$ and the signals are uncorrelated with each other. The two signal powers are equal; the input SNR is 0 dB and the number of snapshots is 200~800, with the interval being 200. The mutual coupling coefficient is consistent with the first experiment.

As shown in Figure 4, the proposed method has the best DOA estimation performance under any snapshot, and as the snapshot increases, the DOA estimation error gradually decreases.

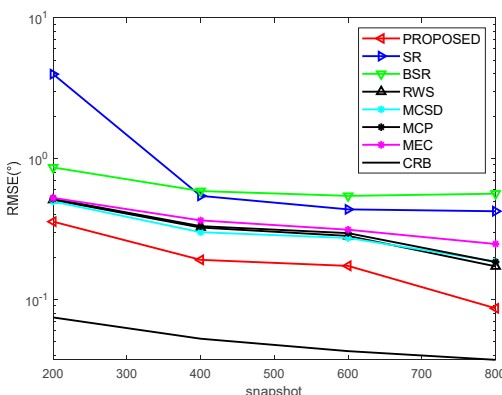

**Figure 4.** The DOA estimation performance with different snapshot.

The fourth experiment is in order to illustrate the advantages of the proposed algorithm in the PR under different SNR: set the simulation parameters as follows: the number of uniform linear array elements M = 9, there are two far-field narrowband signals that are incident from $\theta_1 = -30.1°$ and $\theta_2 = 49°$, respectively, and the signals are uncorrelated with

each other, $Q = 2$. The number of sampling snapshots of the two signals is K = 200, and the input SNR is 0~15 dB, with an interval of 5 dB. PR is defined as:

$$PR = (\sum_{n=1}^{N} P_n(\theta)) / N \tag{29}$$

$P_n(\theta)$ represent whether $\theta_q$ are successfully estimated in the *n*-th Monte Carlo experiment. If the estimation is successful, $P_n(\theta) = 1$, otherwise $P_n(\theta) = 0$. The limiting condition for successful estimation of $P_n(\theta)$ is: any $\hat{\theta}_{n,q}$ satisfies $|\hat{\theta}_{n,q} - \theta_q| \leq 0.5$. The mutual coupling coefficient is consistent with the first experiment.

The experiment sets the success condition for DOA estimation to be that any $\theta_q$ satisfies $|\hat{\theta}_{n,q} - \theta_q| \leq 0.5$. This experiment is to compare the stability of different algorithms. As shown in Figure 5, the input SNR gradually increases, and the PR of all algorithms gradually improves. At the same time, our proposed algorithm has the highest PR under any SNR, so it has the best stability under different SNRs.

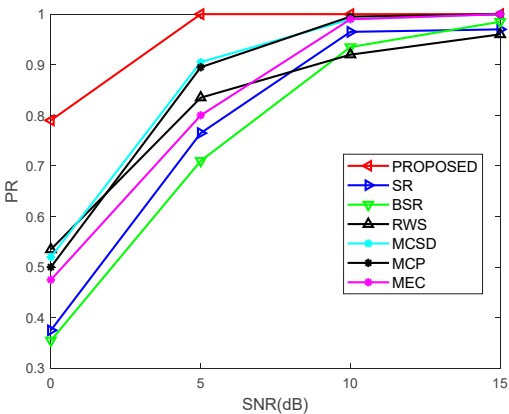

**Figure 5.** The PR with different SNR.

The fifth experiment is in order to illustrate the advantages of the proposed algorithm in the PR under different snapshots: set the simulation parameters as follows: the number of uniform linear array elements M = 9, there are two far-field narrowband signals that are incident from $\theta_1 = -30.1°$ and $\theta_2 = 49°$, respectively, and the signals are uncorrelated with each other, $Q = 2$. The number of sampling snapshots of the two signals is K = 200~800, with an interval of 200, and the input SNR is 0 dB. The mutual coupling coefficient is consistent with the first experiment. The successful condition for DOA estimation is the same as in the fourth experiment.

As shown in Figure 6, as the input snapshot gradually increases, the PR of all algorithms gradually improves. At the same time, our proposed algorithm has the highest PR under any snapshot, so it has the best stability under different snapshots.

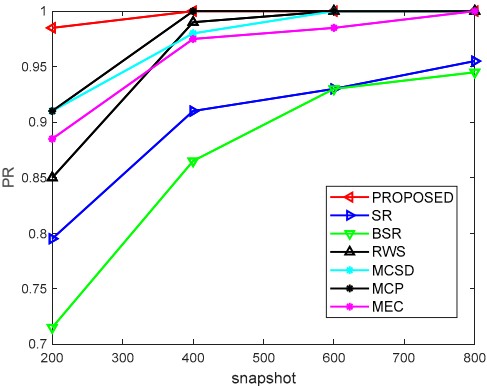

**Figure 6.** The PR with different snapshots.

## 5. Conclusions

In this paper, a novel weighted block sparse method for DOA estimation under unknown mutual coupling is proposed based on signal subspace and utilizing the special deformation of MCM and steering vector product to achieve excellent DOA estimation performance. Firstly, the signal subspace is obtained by decomposing the eigenvalues of the sampling covariance matrix. Then, a block sparse model is established based on the signal subspace. To enhance sparsity, a suitable set of weights is calculated to enhance the performance of DOA estimation. Finally, to facilitate the solution of the optimization problem, it is transformed into a SOC problem. Simulation has demonstrated the performance advantages of the proposed algorithm. It should be noted that the block sparsity method is solved using CVX, and more efficient solving methods still need to be studied in the future. The noise subspace is essential to the proposed method, so it is not suitable for scenarios of coherent signals.

**Author Contributions:** Conceptualization and simulation, Y.L.; Writing—original draft, Y.Y. and H.L.; Writing—review and editing, K.T. All authors have read and agreed to the published version of the manuscript.

**Funding:** This research was funded by Stable-Support Scientific Project of China Research Institute of Radiowave Propagation, grant number A132304295-006.

**Data Availability Statement:** Data are contained within the article.

**Conflicts of Interest:** The authors declare no conflict of interest.

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
