# Peer review of "A Novel Weighted Block Sparse DOA Estimation Based on Signal Subspace under Unknown Mutual Coupling"

_electronics, doi:10.3390/electronics13091790_

Round 1
Reviewer 1 Report
Comments and Suggestions for Authors
In this manuscript, the authors proposed a novel weighted block sparse method based on the signal subspace to realize DoA estimation under unknown mutual coupling in ULAs. The manuscript tackles a very interesting topic. However, I cannot recommend this paper for publication in its present form since technical and language revision is needed. Please refer to the attached PDF for more information.

The paper needs an overall language revision. Please refer to the attached PDF for more information.
Reviewer 2 Report
Comments and Suggestions for Authors
- Would be nice to include matrix dimensions consistently in the publication e.g A is MxQ and so on
- You should go into more detail about the solver used to solve the SOC e.g. ECOS,SCS etc: You have formulated a compressive sensing like problem and chose the SOC using weighted l1-norm since its more robust against noise. This is a well known solving method and you should describe in detail what kind of solver has been used and what has been the sampling grid e.g. what angular resolution have you assumed.
- The paper could benefit from explaining how you handle points between grid-points e.g. by interpolation
-In the evaluation section you should name the comparative approaches rather than writing method[xy]
-You should highlight the strengths and weaknesses of your approach: sparsity assumption need to be valid etc.
-One suggestion for the outlook: You could also use the measured array manifold rather than numerically construct it, so you would directly include the mutual coupling effects.
Comments on the Quality of English Languageminor grammar editing and typos
Reviewer 3 Report
Comments and Suggestions for Authors
[1] Introduction: Please modify the writing style of citing a reference.
[2] Figure 1: Suggest to put a label on each antenna element.
[3] Eqn.(1): Please clarify the composition of X(t) in a vector form.
[4] Line 87: Please elaborate how these c coefficients are selected.
[5] Eqn.(9): Please elaborate the rational of bringing the coupling matrix C in the denominator.
[6] Eqn.(11): Please justify the last equality.
[7] Eqn.(15): Please define “B \times C$.
[8] Eqn.(18): Suggest to add references on “compressive sensing” techniques.
[9] Line 201: Please explain “weighted values”.
[10] Line 231: Please justify the selection of “c=[1, 0.684+0.563i]”.
[11] Figure 3: Please briefly review the other methods for comparison.
[12] Suggest to add CRB (Cramer-Rao bound) to Figures 3 and 4.
[13] Line 287: Please justify the selection of “0.5”
Comments on the Quality of English LanguageGrammar and writing style need to be improved, professional editing service is recommended.
Round 2
Reviewer 1 Report
Comments and Suggestions for Authors
The authors addressed my previous comments. Nevertheless, before recomending the paper for publication, I still have some inputs.
-ll. 63-66 Your claim should be supported by some references, otherwise it might look as your opinion. As I stated in my previous review, please consider expanding the enounced literature by including more recent work on lightweight technique for artifacts correction and compensation in Direction of Arrival Estimation.
-l. 219 Thanks for adding those details on the simulation setup. However, I think you should also add the reasons why you chose such setup (e.g. the low frequency values are due to a preceeding down-conversion stage? What is the initial RF frequency you supposed? Does the frequency impact the overall algorithm complexity and performance?)
Comments on the Quality of English LanguageThe authors addressed my concerns about English writing.
Reviewer 3 Report
Comments and Suggestions for Authors
Previous comments have been addressed.
Comments on the Quality of English LanguageMinor editing can further improve the quality.
